# Effects of Sesquiterpene Lactones on Primary Cilia Formation (Ciliogenesis)

**DOI:** 10.3390/toxins15110632

**Published:** 2023-10-27

**Authors:** Marina Murillo-Pineda, Juan M. Coto-Cid, María Romero, Jesús G. Zorrilla, Nuria Chinchilla, Zahara Medina-Calzada, Rosa M. Varela, Álvaro Juárez-Soto, Francisco A. Macías, Elena Reales

**Affiliations:** 1Research Unit, Biomedical Research and Innovation Institute of Cádiz (INiBICA), Department of Urology, University Hospital of Jerez de la Frontera, 11407 Jerez, Spain; marina.murillo@inibica.es (M.M.-P.); maria.romero@inibica.es (M.R.); zahara.medina@inibica.es (Z.M.-C.); alvaro.juarez.sspa@juntadeandalucia.es (Á.J.-S.); 2Allelopathy Group, Department of Organic Chemistry, Institute of Biomolecules (INBIO), School of Science, University of Cadiz, Campus de Excelencia Internacional (ceiA3), 11510 Puerto Real, Spain; jcoto@us.es (J.M.C.-C.); jesus.zorrilla@uca.es (J.G.Z.); nuria.chinchilla@uca.es (N.C.); rosa.varela@uca.es (R.M.V.); 3Department of Organic Chemistry, University of Seville, 41012 Seville, Spain; 4Department of Chemical Sciences, University of Naples Federico II, Complesso Universitario Monte S. Angelo, Via Cinthia 4, 80126 Naples, Italy

**Keywords:** sesquiterpene lactones, grosheimin, costunolide, primary cilia, ciliogenesis

## Abstract

Sesquiterpene lactones (SLs), plant-derived metabolites with broad spectra of biological effects, including anti-tumor and anti-inflammatory, hold promise for drug development. Primary cilia, organelles extending from cell surfaces, are crucial for sensing and transducing extracellular signals essential for cell differentiation and proliferation. Their life cycle is linked to the cell cycle, as cilia assemble in non-dividing cells of G_0_/G_1_ phases and disassemble before entering mitosis. Abnormalities in both primary cilia (non-motile cilia) and motile cilia structure or function are associated with developmental disorders (ciliopathies), heart disease, and cancer. However, the impact of SLs on primary cilia remains unknown. This study evaluated the effects of selected SLs (grosheimin, costunolide, and three cyclocostunolides) on primary cilia biogenesis and stability in human retinal pigment epithelial (RPE) cells. Confocal fluorescence microscopy was employed to analyze the effects on primary cilia formation (ciliogenesis), primary cilia length, and stability. The effects on cell proliferation were evaluated by flow cytometry. All SLs disrupted primary cilia formation in the early stages of ciliogenesis, irrespective of starvation conditions or cytochalasin-D treatment, with no effect on cilia length or cell cycle progression. Interestingly, grosheimin stabilized and promoted primary cilia formation under cilia homeostasis and elongation treatment conditions. Thus, SLs have potential as novel drugs for ciliopathies and tumor treatment.

## 1. Introduction

The primary cilium (PC) is a single nonmotile organelle that projects from the cell surface in nearly all cell types [1,2]. It is made up of the ciliary membrane that surrounds a microtubule-based structure, the axoneme, which is nucleated from the basal body [3]. Analogous to an antenna, the cilium transduces chemical and mechanical external signals through membrane receptors localized in the ciliary membrane. Signaling pathways dependent on primary cilia include Sonic Hedgehog (Shh), Wingless/Int (WNT), and Transforming Growth Factor-β (TGF-β), which are essential for proper tissue development and cell homeostasis [4]. The presence of cilium is regulated by the cell cycle, as PC emerges on G_0_ (quiescent) or the early G_1_ phase, and it is maintained until cells enter mitosis, being resorbed during G_2_/M transition [5]. The formation of primary cilia (ciliogenesis) in non-polarized cells is a process highly regulated. Briefly, it starts with the attachment of the distal end of the mother centriole to a ciliary vesicle, which is mediated by the centriolar distal appendages. After docking, the ciliary vesicle grows with the axoneme and gives rise to the ciliary sheath, whose fusion with the plasma membrane results in the emergence of the cilium in the extracellular environment [6,7]. Once primary cilium is a mature surface-exposed cilium, the activation of cellular pathways that regulate the ciliary length and cilia assembly and disassembly are activated [5,8]. Thus, the PC is a result of coordinated trafficking, docking, and fusion of vesicle events that will define the unique composition of the ciliary membrane and the membrane domain at the base of the cilia. Many molecular details of these pathways remain unresolved.

Defects in PC and motile cilia function or formation are associated with a growing list of human developmental and degenerative disorders, collectively referred to as ciliopathies, that include Nephronophthisis (NPHP), Retinitis Pigmentosa (RP), Lebel Congenital Amaurosis (LCA), Bardet-Biedl Syndrome (BBS), Joubert Syndrome (JBTS), Primary Ciliary Dyskinesia (PCD), and Polycystic Kidney Disease (PKD) [9,10]. These ciliopathies present with diverse etiologies that include kidney disease, lung disorders, and loss of vision. Additionally, ciliary dysfunction is also associated with several types of cancer. It has been shown that in breast, prostate, and other cancer types, cilia loss is required for tumor progression; instead, in medulloblastoma and basal cell carcinoma, cilia retention or active ciliation is required [11,12]. Highly metastatic astroglioma cells have internal cilium precursors that fail to protrude, which may allow cancer cells to evade regulatory checkpoints [13]. Thus, there is a growing need to find new active compounds able to modify PC function to provide novel strategies for their treatment.

Natural products are important leads in drug discovery. Sesquiterpene lactones (SLs) constitute a large and diverse group of biologically active secondary metabolite plant chemicals [14,15]. They are terpenoids with a basic structure of 15 carbons, and their usual feature is a γ-lactonic ring that can also contain α-methylene groups, hydroxyls, esterified hydroxyls, or epoxides rings [16]. SLs and their derivatives show a promising role in drug development as they have promising anticancer and anti-inflammatory effects, antifungal, analgesic, antimalarial, and antimicrobial activities, among others, or they can be used in combination therapy as sensitizing agents to enhance the action of drugs in clinical use [15,17]. There are several structure–activity relationship analyses showing that several SLs react with thiols via rapid Michael-type addition. These reactions are mediated by the α- and β-unsaturated carbonyl systems and depend on the geometry of the molecule, although additionally, other factors such as lipophilicity and chemical environment may also influence the activity [18]. Although it has been described that SLs possess a broad spectrum of biological activities, the effect on PC formation or function has not been studied yet. Here, we report the capacity of the SLs grosheimin, costunolide, and α-, β-, and γ-cyclocostunolide to perturb primary cilium biogenesis in human retinal pigment epithelial (RPE) cells, a model of non-polarized retinal cells. Ciliogenesis can be induced by cell starvation during 24 h or accelerated by actin destabilization via cytochalasin D treatment. We observed a reduction in cilia percentage in RPE cells after treatment with all products tested under both cilia induction conditions without affecting cell cycle progression or cilia length. Interestingly, grosheimin showed an increase in cilia formation at the late stages of ciliogenesis, where cell pathways for cilia stabilization are activated. Altogether, we propose the potential of SLs to be investigated as drugs for the treatment of ciliopathies and/or tumors.

## 2. Results and Discussion

From a small library of plant allelochemicals synthesized previously in our lab and based on previous biological studies in human cells [19], two sesquiterpene lactones were selected to elucidate their function in cilia formation: grosheimin [20] isolated from a leaf extract from *Cynara scolymus*, and costunolide [21] isolated from a root extract from *Saussurea lappa*. From the former, three eudesmanolide-type sesquiterpene lactones were also obtained by cyclization reaction using *p*-TsOH (para-toluenesulfonic acid), generating three cyclocostunolides with a characteristic double bound located in different positions (α,β,γ) [22] (Figure 1), which are likewise natural products isolated from plants [23,24].

### 2.1. Effect of Sesquiterpene Lactones on Cell Viability and Cell Cycle Progression

First, we evaluated the cytotoxicity of the products in telomerase-immortalized retinal pigment epithelial (RPE-1) cells, a well-known human cell line used to study primary cilia formation in non-polarized cells [25,26]. For determining the cell viability after treatment with the compounds, we used the crystal violet assay [27], a simple and widely used method to assess the impact of molecules on cell survival and growth inhibition. As crystal violet dye binds to proteins and DNA, it stains only living cells that are attached in the culture, and we can obtain the % of surviving cells under diverse stimulated conditions. RPE cells at high confluency (90–100%) were treated with SLs ranging from 1 to 25 μM final concentration for 24 h in a serum-free media, and cell viability was scored using the crystal violet assay. Treatment with solvent 0.1% DMSO was used as a control. All compounds tested showed no effect on cell viability at 10 μM concentration or below. Grosheimin (G), costunolide (C), and β-cyclocostunolide (β-C) were also innocuous at 25 μM, while α- and γ-cyclocostunolides (α-C; γ-C) showed around 60% viability decrease (Figure 2A). Most products displayed toxicity in RPE cells at concentrations exceeding 50 μM. For the rest of the experiments, 10 μM final concentration was selected to address effects on cell cycle and primary cilia formation, in order to maintain the highest cell viability in our ciliogenesis assays.

The primary cilia life cycle is tightly coupled with the cell cycle. Cilia assemble in non-dividing quiescent or post-mitotic differentiated cells (G0_0_/G_1_ phases) and disassemble before entering mitosis [28]. Although the connection between the cell cycle and primary cilia is still under exploration, there appears to be a bidirectional crosstalk between cilium formation and cell division, as improper division can result in abnormal cilia formation and failure to form a cilium can regulate the cell cycle. For example, over-proliferative cancer cell lines generally lack cilia, and cells that cannot properly form a cilium undergo inappropriate cell division [29]. In addition, several studies have documented changes in cell cycle progression in the context of cancer cells under the effect of some SLs in the search for therapeutic anti-tumoral treatments. Such is the documented case for alantolactone or eremanthin, compounds that affect cell cycle progression, arresting cancer cells at the G_2_ phase [30,31].

To test if the selected SLs have an effect on cell cycle progression in non-tumoral cells, we analyzed the cell cycle status of RPE cells treated with the test compounds for 24 h by flow cytometry analysis (Figure 2B). Cells were stained with the intercalating DNA dye propidium iodide, which allows the detection of cellular DNA content. Via flow cytometry, the DNA staining generates a histogram of fluorescence intensity that represents the distribution of cells in the different phases of the cell cycle (G_0_/G_1_, S, and G_2_/M). Cell cycle analysis showed that the treatment with all the compounds tested in RPE cells during 24 h does not affect cell cycle progression. As compounds were added in a starvation medium which inhibits cell proliferation, cells were mostly arrested in the G_1_ phase. These data indicate that incubation with the SLs grosheimin (G), costunolide (C), and α-, β-, and γ-cyclocostunolide at 10 μM does not perturb the cell cycle progression at tested conditions. The fact that the SLs under study in this work do not affect cell cycle progression suggests that any perturbation over ciliogenesis detected would be independent from the cell cycle.

### 2.2. Effect of Sesquiterpene Lactones on Early Steps of Primary Cilia Formation

PC formation in RPE cells can be induced by the absence of serum in the cell culture for at least 24–48 h [32]. Via live cell imaging, it has been described that newly forming and elongating internal cilia happen around the first 2–4 h after inducing cilia formation by starvation [25]. Once internal cilia are formed, it will fuse with the plasma membrane. Mature external primary cilia elongation and composition are maintained via different membrane trafficking pathways in the cell until it is disassembled to start a new cell cycle [33]. To determine the effect of the selected SLs on the early steps of ciliogenesis, RPE cells were serum starved by incubation for 24 h with a medium containing 0.5% serum to induce cilia formation, and tested compounds were added at the same time at 10 μM final concentration. PC (percentage and length) was analyzed using confocal immunofluorescence microscopy. Acetylated α-tubulin and γ-tubulin were used as a ciliary marker and centrosome marker, respectively. DAPI (4′,6-diamidino-2-phenylindole) fluorescent stain, which binds to DNA, was used for nucleus labeling. The addition of all SLs further affected early cilia formation in RPE cells. Grosheimin was the one with a percentage of primary cilia closer to control levels, and costunolide, β-cyclocostunolide and α-cyclocostunolide were the most dramatic ones showing less than 5% ciliated cells (Figure 3A,B). γ-cyclocostunolide showed a bimodal response as sometimes cells were similar to control levels (0.1% DMSO), and sometimes cells were not ciliated at all and even showed some DAPI phenotype indicative of cell damage (Figure 3A,B). We also measured the length of primary cilia, and there were no significant differences compared to the control condition for all tested compounds (Figure 3B), indicating that although cells treated with SLs can ciliate less in serum starvation conditions, the cells that manage to ciliate produce a normal-length cilium. No differences were observed for the centrosome labeling, except for cells treated with γ-cyclocostunolide, in which the centrosome has dispersed labeling, probably associated with cell damage or cytoskeleton damage. Further studies need to be conducted to know the reason for this phenotype.

Together, these findings described that the natural compounds grosheimin (G), costunolide (C), and α-, β-, and γ-cyclocostunolide are able to interrupt cilia formation in non-polarized and non-tumoral cells, without affecting the cell cycle progression (as shown in Figure 2B). Further work will be important to investigate the molecular target of the tested SLs and/or signaling pathways affected.

### 2.3. Effect of Sesquiterpene Lactones on Primary Cilia Formation Induced by Cytochalasin D

PC can also be induced by the actin polymerization inhibitor cytochalasin D (CytD), which facilitates ciliogenesis and promotes cilium elongation independently of serum starvation [32,34]. Cytochalasin D induces ciliogenesis at doses that do not affect stress fiber formation, excluding the possibility of global actin cytoskeleton rearrangement in ciliogenesis control. To examine the effect of the selected SLs on ciliogenesis induced by CytD and on actin dynamics, we treated RPE cells with CytD for 18 h together with the SLs grosheimin, costunolide, or α-, β-, and γ-cyclocostunolide. As expected, the addition of CytD and DMSO at the same time increased ciliated cells compared to DMSO-only treated cells (Figure 4A,B). The addition of all the SLs further affected cilia formation in a similar way as cilia-induced starvation conditions (Figure 3). Grosheimin showed less effect, while all the other compounds showed a decrease in the ratio to 0.4–0.5 (Figure 4B). The variability associated with γ-cyclocostunolide was not detected in this treatment, pointing to a cellular defect due to the addition of the compound under starvation conditions. We further measured the length of primary cilia, and we observed that there were no significant differences compared to the control condition (CytD and DMSO) for all tested compounds (Figure 4B). We also analyzed the cell cycle progression in CytD treatment conditions in RPE cells. It has been reported that CytD treatment inhibits the cell cycle progression from G_0_ to S phase and also mitosis during cytokinesis [35,36]. Therefore, treatment of RPE cells with CytD for 18 h (control) showed an increase in the population with DNA content = 2, as cytokinesis blockage leads to the accumulation of tetraploid cells that in G_1_ will have the same amount of DNA as diploid G_2_ cells. Nevertheless, incubations with the allelochemicals plus CytD resulted in no effect on the profile compared to the control condition (Figure 4C).

Blocking actin assembly facilitates ciliogenesis by stabilizing a pericentrosomal preciliary compartment (PPC), a transient tubular and vesicular compartment in charge of sorting transmembrane proteins destined for cilia during the early ciliogenesis. It is observed after initiation of ciliogenesis (4 h after starvation) and disappears after 24 h serum starvation from the ciliary base. Previous studies have shown that CytD-treated cells promote PPC formation (in 2 h), facilitating both axoneme assembly and ciliary membrane biogenesis [32]. Because SLs block ciliogenesis induced by CytD and starvation, it is therefore tempting to speculate that the allelochemicals tested in this study may interfere in the formation and stabilization of the preciliary compartment, a process that involves fusion of transport of vesicles at the base of the cilia, via the recruitment of lipid and membrane proteins that enables the biogenesis of ciliary membrane and axoneme assembly.

### 2.4. Effect of Sesquiterpene Lactones on Late Steps of Ciliogenesis

Ciliogenesis requires enormous coordination of cell cycle regulatory signaling and the recruitment of ciliary proteins with proper stoichiometry. The final step of ciliogenesis is the extension of the ciliary axoneme and ciliary membrane. Its regulation is orchestrated by intracellular trafficking, intraflagellar transport (IFT), and autophagy, among others [37,38]. Because primary cilia are formed in quiescent cells and resorbed during the G_2_/M transition, the coordination of these two processes must be highly robust. It has been shown that ciliary resorption is related to stress responses [39], cell cycle progression, and cell differentiation [28]. To see if SLs have an effect in maintaining the equilibrium between cilia assembly and disassembly and cilia extension, we also analyzed the effect of the compounds on the late phases of ciliogenesis, where primary cilia are already facing the extracellular media, axoneme is assembled, and ciliary membrane formed. Cells were serum-starved for 24 h to induce primary cilia formation, and compounds were added for an additional 24 h in the serum-starved medium. In these conditions, the grosheimin compound showed an increase in the ratio of total ciliated cells and α-cyclocostunolide and γ-cyclocostunolide a decrease (Figure 5A,B). For the latter products, α-cyclocostunolide and γ-cyclocostunolide, the reduction in cilia percentage was more than half compared to control. Cilia length was also evaluated (Figure 5B), and α-cyclocostunolide showed a reduction in the cilia length, while grosheimin, costunolide, and β-cyclocostunolide showed an increase, and γ-cyclocostunolide showed any difference. For γ-cyclocostunolide, we again detected significant differences among replicates ranging from no difference in cilia percentage to almost a 35% decrease (Figure 5B), indicating variability when cells are under the action of this product. The reason for this variability needs to be further studied.

The increase in the ciliated cells ratio and ciliary length found in cells treated with grosheimin at late stages of ciliogenesis, together with the decrease in percentage detected at early steps, suggests that grosheimin has a positive effect on the assembly and/or length of the primary cilia, promoting a higher rate of elongation and ciliary-assembly pathways than ciliary-disassembly ones. Given that grosheimin incubation does not affect cell viability or cell cycle progression, this could be a promising compound to specifically promote ciliogenesis. It could be studied in ciliopathies that result from abnormally short cilia or impaired ciliogenesis, such as Bardet-Biedl syndrome (BBS), Nephronophthisis (NPHP), Meckel syndrome (MKS), Joubert syndrome (JBTS), short-rib polydactyly syndrome, and cranioectodermal dysplasia syndrome [9]. Although investigations in primary cilia and disease have been conducted mainly in ciliopathies, some documented changes in ciliation have also been shown in cancer initiation and progression. Differences in ciliation between cancer cells and cells from the tumor microenvironment contribute to the growth of the tumor. In some cancer types, a loss of ciliation promotes oncogenesis and cancer-related signaling [40,41]. Cilia loss occurs during the early stages of breast, prostate, pancreatic, and other cancer types [29]. Knockdown of the gene encoding tubulin monoglycylase (TTLL3), which reduces the number of primary cilia and increases the proliferation of colon epithelial cells, strongly promotes the development of colorectal carcinomas in a mouse model [42]. In these types of tumors, grosheimin could be considered being studied as a new drug for treating tumors, increasing ciliogenesis, and reducing cell proliferation.

α-cyclocostunolide showed a reduction in both the ciliated cell ratio and cilia length compared to the control condition. This indicates a strong effect on the cilia elongation process and makes this compound an interesting tool to test in ciliopathy models such as asphyxiating thoracic dystrophy, where increased ciliogenesis is observed, as well as ciliopathies characterized by excessively long cilia such as juvenile cystic kidney disease [43] and tuberous sclerosis [44]. Following the same idea, treatment with α-cyclocostunolide that shows opposite effects might be interesting in conditions where cilia reduction is needed. For example, Sonic Hedgehog (SHH) signaling at primary cilia drives the proliferation and progression of a subset of medulloblastomas [45].

## 3. Conclusions

The major function of primary cilia is to sense a variety of external stimuli, including flow, ligands, and light, to regulate cell homeostasis, proliferation, and differentiation. Ciliation varies during organismal development, and cells oscillate between ciliated and non-ciliated stages in cycling cells, where the presence of cilia antagonizes cell cycle progression [37]. Abnormalities in ciliogenesis result in cancer, pleiotropic disorders, and denominated ciliopathies, and no treatment options are currently available for the latter. In this study, we demonstrate for the first time that the sesquiterpene lactones grosheimin, costunolide, and α-, β-, and γ-cyclocostunolide can modify primary cilium formation in a cellular model of human normal non-polarized cells, RPE. The activity exhibited by SLs serves as a promising and expanding strategy for the treatment of diseases related to primary cilium dysfunction.

We propose that grosheimin, costunolide, and α-, β-, and γ-cyclocostunolide are able to block ciliogenesis via targeting proteins that are implicated in the early stages of ciliogenesis, where trafficking and docking of ciliary membranes to the mother centriole happens (Figure 6). Growing evidence indicates that some oncogenic signaling pathways induce ciliation while others repress it. Finding new chemicals capable of blocking cilia formation would be useful in tumors where tumor cell growth is dependent on ciliary signaling. The new function of SLs on cilia formation, shown in this work, encourages further studies on the development of new anticancer drugs with a γ-lactone ring that had primary cilia as a target. Further work is needed to identify the SL targets and effects on tumor cells.

Interestingly, at the late steps of ciliogenesis, where axoneme and ciliary membrane stabilization and elongation happens, the effect of grosheimin is the opposite, promoting ciliogenesis and elongation of primary cilium in normal cells (Figure 6). This suggests a specific effect of grosheimin in promoting cilia assembly against disassembly, pointing out a cellular function specific to the molecule, perhaps related to the -thiols group on its structure. Primary cilium dysfunction diseases exhibit little genotype–phenotype relationship. Mutations in a single gene can be implicated in multiple distinct ciliopathies affecting multiple organs with variable manifestations. Compounds that target disease-associated symptoms or cilia biogenesis are developed as an alternative therapeutic approach. This preliminary result showing that grosheimin promotes cilia assembly encourages further studies with this compound to be tested in cellular models of disease and to be studied as a therapy to treat ciliopathies and tumor growth.

Ciliopathies and diseases exhibiting affected ciliary function are increasing, and identifying modulators of ciliogenesis to connect ciliogenesis with other basic cellular processes assumes increasing importance. Much more work is required to address SL molecular targets and solubility in order to increase our knowledge of primary cilium function and structure and create new therapies for ciliary dysfunction diseases, an emerging field of study.

## 4. Materials and Methods

### 4.1. Plant Material and Extraction and Isolation of Test Compounds

Reactions were quality assessed via analytical thin layer chromatography on pre-coated silica gel 60 F254 glass plates (Merck, Rahway, NJ, USA) in the normal phase and 60 RP-18 F254S (Merck) in the reverse phase. Compound detection was carried out via exposition to UV (λ = 254 nm) using oil (CH_3_COOH: H_2_O: H_2_SO_4_ 4:20:1) and thermic treatment (250 °C).

Product purification was conducted via (i) column chromatography with silica gel Merck 60 (pore size 0.063–0.0200 mm, 0.040–0.063 mm) using gravity elution with the indicated compounds; (ii) semi-preparative HPLC using LiChrospher (Torrance, CA, USA) 100 RP-18 (10 μm) in reverse phase in Merck-Hitachi with refraction index detector equipment; and (iii) solid phase extraction (SPE) using StraraTM-X (Não-Me-Toque, Brazil) (33 µm Polimeric Reverse Phase, 200 mg/3 mL). Organic solvents and reagents were obtained from Fisher Chemical^®^ (Waltham, MA, USA), VWR, Panreac^®^ (Castellar del Vallès, Spain), and Sigma-Aldrich^®^ (St. Louis, MO, USA). Synthesized compounds were identified using NMR spectra in chloroform-d using AGILENT^®^-500 MHz equipment (Santa Clara, CA, USA). Spectra obtained were one-dimensional ^1^H-RMN and ^13^C-RMN and two-dimensional ^1^H-COSY (^1^H-^1^H correlation) and HSQC (^1^H-^13^C correlation).

**Costunolide** ((3a*S*,6*E*,10*E*,11a*R*)-6,10-dimethyl-3-methylene-3a,4,5,8,9,11a-hexahydrocyclodeca[*b*]furan-2(3*H*)-one) was isolated from a root extract from *Saussurea lappa* purchased from Pierre Chauvet S.A. (Seillans, France). A total of 50 g of *S. lappa* root extract was dissolved in CH_2_Cl_2_ and fractioned by column chromatography using *n*-hexane (0.6 mL) as first eluent, and then a mixture of *n*-hexane/AcOEt 95:5 (6 L) [46]. Those fractions containing costunolide, according to thin-layer chromatography analysis, were further purified by column chromatography using *n*-hexane/AcOEt 95:5 to obtain costunolide in a total amount of 2.3 g (4.6%). Spectroscopy data were in agreement with those already reported for costunolide [47].

**Grosheimin** ((3a*S*,6a*R*,9*S*,9a*R*,9b*R*)-9-methyl-3,6-dimethyleneoctahydroazuleno[4,5-*b*]furan-2,8(3*H*,4*H*)-dione) was isolated from a leaf extract from Cynara scolymus. An amount of 12.5 g of C. scolymus leaf extract was dissolved in EtOAc. Colum chromatography separation with hexane/acetone 70:30 to 20:80 was performed and the eluted fraction containing the lactones was separated again using n-hexane/acetone 60:40. The fraction were monitored with analytical thin layer chromatography using H_2_O:acetone 60:40 and the fraction with less polar products was extracted by SPE using H_2_O:acetone 35% obtaining grosheimin (9.2 mg and 0.07%). Spectroscopy data were in agreement with those already reported for grosheimin [48].

### 4.2. Semi-Synthesis

Isolated costunolide (120 mg, 0.517 mmol) was stirred in CH_2_Cl_2_ (5 mL), and p-TsOH (para-toluenesulfonic acid; 24.3 mg, 0.141 mmol) was added for cyclization reaction at room temperature for 4 h [49]. The crude mixture was diluted with CH_2_Cl_2_ (20 mL) and treated with NaHCO_3_-saturated aqueous solution (25 mL). The organic phase was separated, and the aqueous phase was washed with CH_2_Cl_2_ (3 × 25 mL). The organic phases were pulled together for neutralization with the NaCl aqueous solution (100 mL). The organic phase was separated again, and aqueous residues were removed with Na_2_SO_4_ addition. The solution was gravity-filtered and concentrated under reduced pressure before purification by column chromatography. The eluant used was n-hexane/AcOEt 95:5. Products by elution order are as follows: γ-cyclocostunolide (8.2 mg, 7% yield: (3a*S*,5a*R*,9b*S*)-5a,9-dimethyl-3-methylene-3a,4,5,5a,6,7,8,9b-octahydronaphtho[1,2-*b*]furan-2(3*H*)-one)), α-cyclocostunolide (42.0 mg, 35% yield: (3a*S*,5a*R*,9b*S*)-5a,9-dimethyl-3-methylene-3a,4,5,5a,6,7,9a,9b-octahydronaphtho[1,2-*b*]furan-2(3*H*)-one)), and β-cyclocostunolide (65.9 mg, 55% yield: (3a*S*,5a*R*,9b*S*)-5a-methyl-3,9-dimethylenedecahydronaphtho[1,2-*b*]furan-2(3*H*)-one)). Spectroscopy data were in agreement with those already reported for α-, β-, and γ-cyclocostunolide [49].

### 4.3. Cell Culture Conditions and Biological Assays

The human cell line used was hTERT-immortalized retinal pigment epithelial cells (hTERT RPE-1) provided by Dr. Fernando Balestra at Cabimer, Seville, Spain. Cells were grown at 37 °C, 5% CO_2_ in DMEM/F-12 containing L-glutamine and 15 mM HEPES cell culture media. Media was supplemented with 10% FBS, 100 U/mL penicillin, 100 µg/mL streptomycin, and 3 µg/mL ciprofloxacin (NORMON laboratories).

For cilia experiments, cells were seeded at a density of 70,000 cells/well in 12-well plates, and the following day, cilia were induced by incubating cells for 24 h with media supplemented only with 0.5% FBS. In experiments aimed to see the effect of test compounds in cilia formation, test compounds were added at the same time as the 0.5% FBS media at a final concentration of 10 µM. In experiments aimed to see the effect of test compounds in established cilia, after incubating cells for 24 h in 0.5% FBS media, test compounds were added for an additional 24 h in 0.5% FBS media at a final concentration of 10 µM. For all experiments, DMSO was added as the control condition (as test compounds are diluted in DMSO stock solution). For cytochalasin D treatment, cells were seeded as above, and the following day, cilia were induced by incubating cells for 18 h with 200 nM cytochalasin D. Test compounds were added at the same time at a final concentration of 10 µM. DMSO was added as the control condition. For cell viability, cells were seeded at a density of 15,000 cells/well in 96-well plates, and the following day, media was changed to media supplemented only with 0.5% FBS and test compounds at different final concentrations or DMSO as a control. After incubating for 24 h, cells were washed with 1X PBS and fixed with 1% glutaraldehyde (in 1X PBS) for 20–30 min. Then, after another 1X PBS wash, crystal violet 0.1% (in H_2_O) solution was added for another 20–30 min. Finally, after one wash with H_2_O, plates were left to dry. To quantify % of cell viability, stained fixed cells were resuspended with 10% acetic acid (CH_3_COOH) solution, and absorbance was measured at 590 nm in a plate reader (sometimes samples were additionally diluted 1/10 in 10% acetic acid to obtain values within the range of the plate reader). Cell viability was calculated as a percentage of the viability of control cells treated with DMSO.

### 4.4. Immunofluorescence (IF)

Cells were seeded onto coverslips in 12-well plates, and cilia experiments were performed as indicated above. For IF, coverslips were fixed with −20 °C cold methanol for 10 min at −20 °C, washed with 1X PBS, and blocked for 1h at RT in IF blocking buffer (5% [*wt*/*vol*] BSA, 0.05% [*vol*/*vol*] Tween in 1X PBS). Cells were incubated with primary antibody diluted in IF blocking buffer overnight at 4 °C. Coverslips were washed three times for 5 min at room temperature in 1X PBS containing 0.05% (*vol*/*vol*) Tween and incubated with secondary fluorescent antibody diluted in blocking buffer for 1h at RT. Coverslips were washed three times again, incubated with DAPI (Sigma), washed with 1X PBS, and mounted in the glycerol-based 2.5% [*wt*/*vol*] PVA-DABCO mounting medium for imaging. Primary antibodies used for staining: γ-tubulin (at concentration 1:1000) from Sigma T3559 and acetylated-tubulin (1:2000) from Sigma T7451; Secondary antibodies: Alexa (San Fancisco, CA, USA) 488 conjugated anti-mouse A32723 from Invitrogen (Waltham, MA, USA) and Alexa 568 conjugated anti-rabbit A11011 from Invitrogen. Both were used at 1:500. DAPI was used as a final concentration of 5 μg/mL for 5 min.

### 4.5. Microscopy

Images were collected at room temperature on a Zeiss (Oberkochen, Germany) LSM 900 inverted confocal microscope using a 40X 1.3 NA oil-immersion objective controlled with Zeiss ZEN software 3.2. Images were collected as 0.5-µm z sections. Images presented in the figures are maximum intensity projections. Fluorophores imaged are those conjugated to secondary antibodies listed above. Images were z-stacked and changed to jpg format using Fiji software 1.53c [27] and subsequently analyzed for cilia proportion and cilia length both manually using Fiji software 1.53c and semi-automatized ACDC software_v0.93 [28] run in MATLAB R2016b (Windows OS). A >4 times magnification of the image is also included for better visualization of the cilia and/or centrosome.

### 4.6. Cell Cycle Analysis and Flow Cytometry

For cell cycle analysis, after the indicated treatments, cells were collected and fixed in cold 70% ethanol for at least 30 min at 4 °C. Then, cells were centrifuged, washed with 1X PBS, centrifuged, and treated with RNAse in 1X PBS at a final concentration of 100 µg/mL for at least 1 h at 37 °C. Finally, propidium iodide was added at a final concentration of 50 µg/mL and incubated, protected from light for at least 1 h before cytometry. Cells were analyzed in a BD FACS Celesta SORP (Franklin Lakes, NJ, USA) with Diva software 9.0. Forward scatter (FS) and side scatter (SS) were used to identify single cells, and PI was visualized using the yellow/green laser with excitation at 561 nm (PI maximum emission 605 nm). Cell cycle analysis was performed using the cell cycle tool of FlowJo software v10.8.1.

### 4.7. Statistical Analysis

All experiments were performed at least in triplicates (when *n* > 3, it is indicated in figure legend), and in each experiment for each condition, around 100 cells were analyzed. Data are expressed as a ratio between the mean values compared to the DMSO control and the standard deviation of the mean (SEM). Data statistics were analyzed in GraphPad Prism. We used Mann–Whitney test analysis to identify any difference between two means (test condition and control) that is greater than the expected standard error. For cell viability experiments, we performed a two-way ANOVA analysis, and for cell cycle experiments, a multiple *t*-test analysis comparing G_1_-S-G_2_ values for each test condition versus control. The *p* values are indicated within the figures.

## Figures and Tables

**Figure 1 toxins-15-00632-f001:**
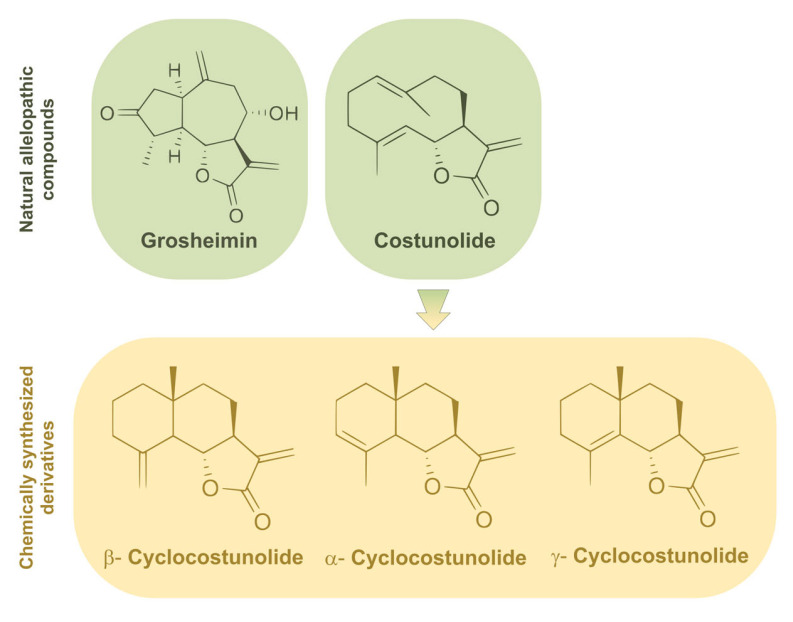
Test compounds used in this study. Natural sesquiterpene lactones isolated from plants are indicated at the top. From costunolide, the three plant-derived eudesmanolides shown at the bottom were synthetized.

**Figure 2 toxins-15-00632-f002:**
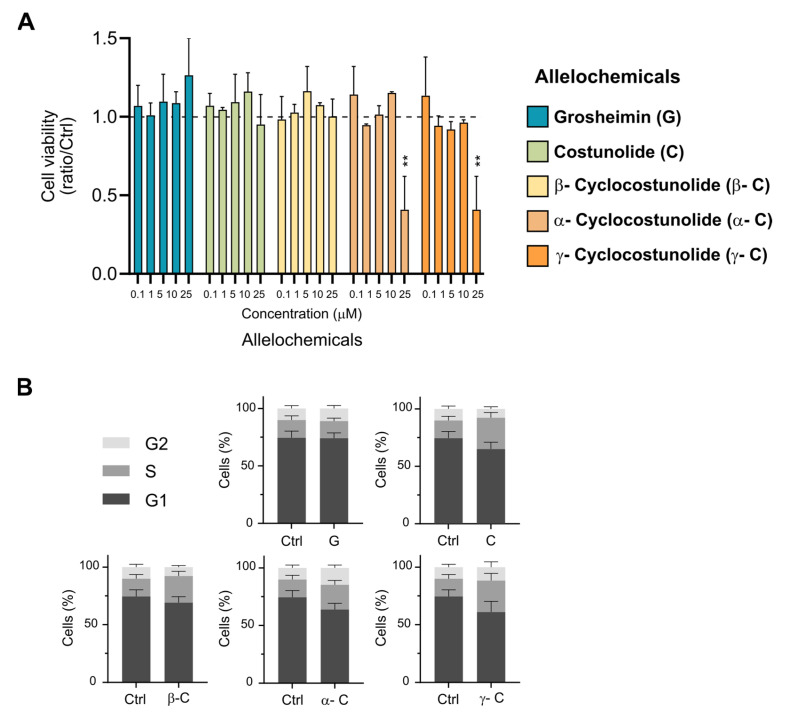
Effect of sesquiterpene lactones on cell viability and cell cycle progression in RPE cell line. (**A**) Effect of increasing concentration of the SLs on viability measured by crystal violet assay. On the right, color code and abbreviation panel indicating selected allelochemicals. Graph shows mean and SEM of 3 independent experiments of the viability ratio compared to control (DMSO-solvent 0.1%). Two-way ANOVA was performed, and significant differences are indicated when *p* < 0.01 (**). (**B**) SLs’ effect on cell cycle progression in cells treated with products at 10 μM final concentration for 24 h in medium containing 0.5% serum. Propidium iodide-stained cells were analyzed by flow cytometry. Graph shows mean and SEM of 3 independent experiments. Multiple *t*-test analysis was performed, and no significant differences were detected.

**Figure 3 toxins-15-00632-f003:**
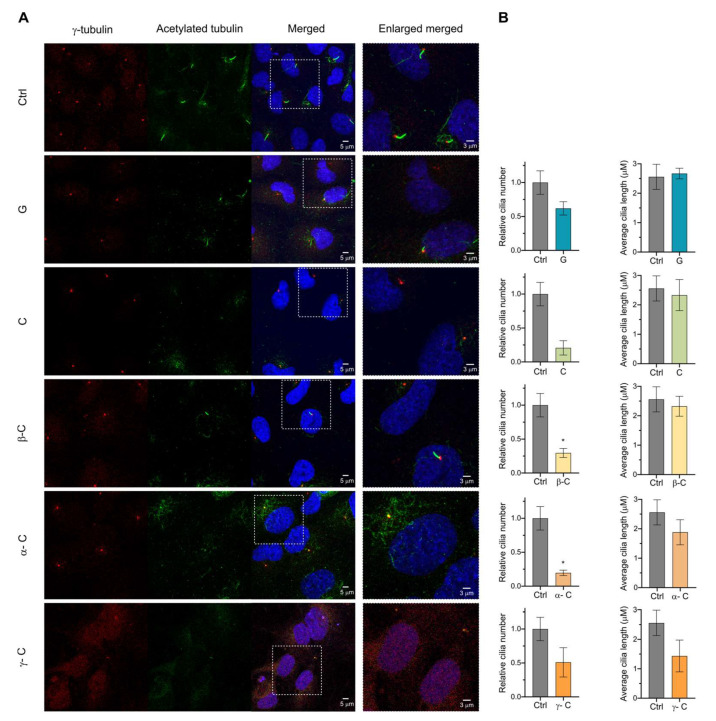
Effect of sesquiterpene lactones on primary cilia formation on RPE cell line. (**A**,**B**) RPE cells were incubated for 24 h with a medium containing 0.5% serum to induce cilia formation, and compounds at 10 μM final concentration were added at the same time. Ctrl: 0.1% DMSO. Immunofluorescence was performed using acetylated α-tubulin to label primary cilia and γ-tubulin to visualize the centrosome. Representative images for each condition are shown (**A**). Graph shows mean and SEM of 4 independent experiments of the ratio of ciliated cells and the average cilia length (**B**). Mann–Whitney test analysis was performed, and significant differences are indicated when *p* < 0.05 (*).

**Figure 4 toxins-15-00632-f004:**
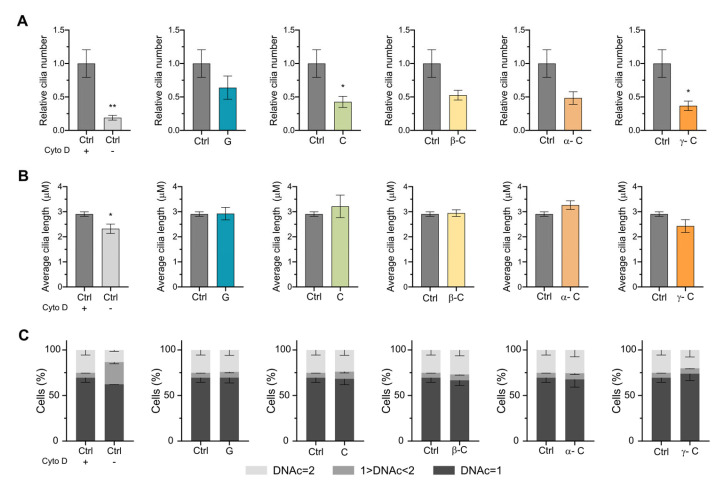
Effect of sesquiterpene lactones on primary cilia induced by cytochalasin D. (**A**–**C**) RPE cells were incubated for 18 h with cytochalasin D at 200 nM final concentration to induce primary cilia formation. Compounds at 10 μM final concentration were added at the same time. (**A**,**B**) Immunofluorescence was performed as in Figure 2. Graph shows mean and SEM of 6 independent experiments of the ratio of ciliated cells (**A**) or the average cilia length (**B**). Mann–Whitney test analysis was performed, and significant differences are indicated when *p* < 0.05 (*) or 0.01 (**). (**C**) Sesquiterpene lactones’ effect on cell cycle under cytochalasin D treatment. Propidium iodide-stained cells were analyzed by flow cytometry. Graph shows mean and SEM of 3 independent experiments. Cytochalasin D blocks cytokinesis after chromosome divergence in anaphase of mitosis; therefore, cell cycle stage and DNAc are not unequivocally linked (ex. DNAc = 2 will contain G_2_ and mitotic cells but also tetraploid cells that will be entering G_1_).

**Figure 5 toxins-15-00632-f005:**
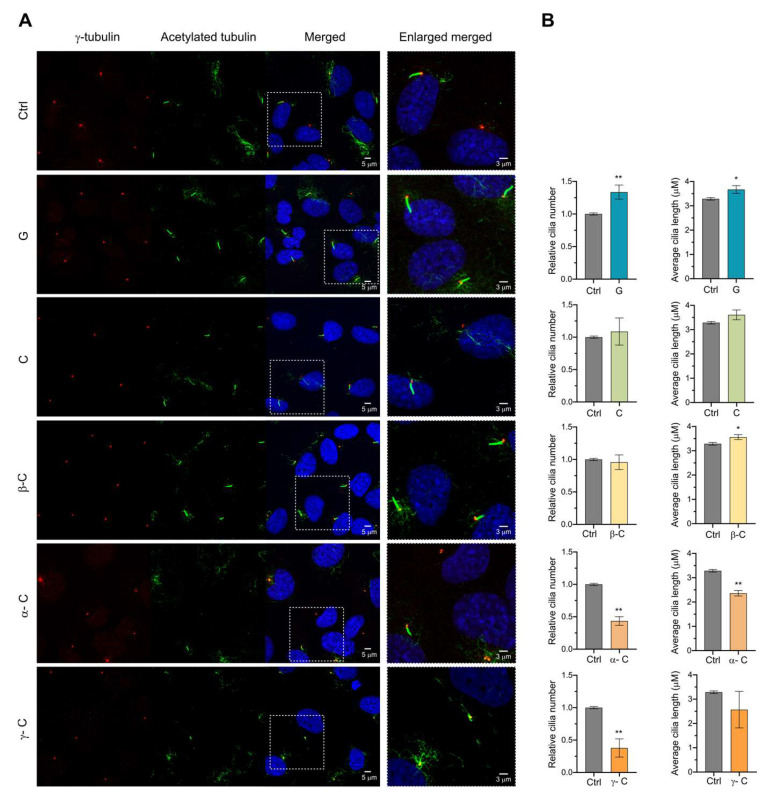
Effect of sesquiterpene lactones on assembled primary cilia. (**A**,**B**) RPE cells were incubated for 24 h in a medium containing 0.5% serum to induce primary cilia formation; then, compounds at 10 μM final concentration were added for an additional 24 h in a medium still containing 0.5% serum. Immunofluorescence was performed as in Figure 3. Representative images for each condition are shown (**A**). Graph shows mean and SEM of 5 independent experiments of the ratio of ciliated cells and the average cilia length (**B**). Mann–Whitney test analysis was performed, and significant differences are indicated when *p* < 0.05 (*), 0.01 (**).

**Figure 6 toxins-15-00632-f006:**
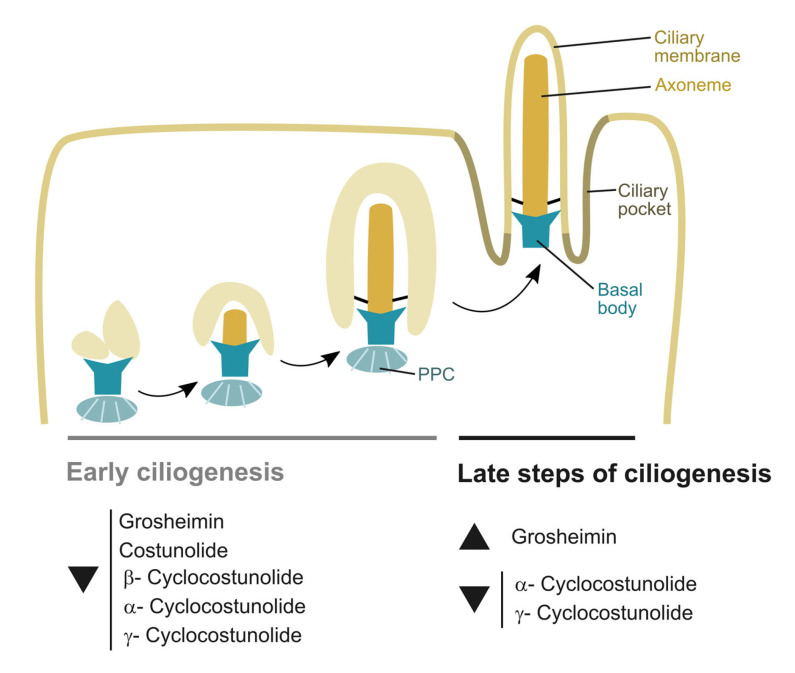
Model of action of SLs on ciliogenesis. The image depicts the SLs that decrease or increase the process of ciliogenesis in RPE cells after 24 h treatment. In early ciliogenesis, products are added at the same time as primary cilia induction by serum starvation or CytD treatment. In the late steps of ciliogenesis, products are added to the cell culture after 24 h of cilia induction by serum starvation.

## Data Availability

Not applicable.

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
