# Peer review of "Effects of Sesquiterpene Lactones on Primary Cilia Formation (Ciliogenesis)"

_toxins, 2023, doi:10.3390/toxins15110632_

Round 1

Reviewer 1 Report

Comments and Suggestions for Authors

The research topic of this article is currently relevant in the scientific community. It considers a promising direction for studying the pathological mechanisms of many human diseases – ciliogenesis. In addition, it is important to note the discovery of new biological effects of sesquiterpene lactones, which represent a large and diverse group of biologically active secondary metabolites of plants. The article is written in good scientific language and will undoubtedly be of interest to the readers of the journal "Toxins" MDPI.

Minor revision:

- In the graph of Figure 4a, cytochalasine-D and 4c have no error value in all samples.
The same is noted for the control group in Figure 3a.
 - The statistical method is not quite correctly chosen.
The Mann-Whitney criterion is used to compare two independent small samples (n ≥ 3).
In this case, it is better to change to a single-factor analysis of variance (ANOVA)
or the Kraskel-Wallis criterion (checking the equality of medians).
 - If the studied compounds affect the primary cilia,
do they cause a change in the structure of microtubules,
which are contained in large quantities in the primary cilia?

Author Response

Reviewer 1:

We appreciate the comments of reviewer 1 and we will be supplying additional information in the revised manuscript to address the suggestions.

Comments and Suggestions for Authors

The research topic of this article is currently relevant in the scientific community. It considers a promising direction for studying the pathological mechanisms of many human diseases – ciliogenesis. In addition, it is important to note the discovery of new biological effects of sesquiterpene lactones, which represent a large and diverse group of biologically active secondary metabolites of plants. The article is written in good scientific language and will undoubtedly be of interest to the readers of the journal "Toxins" MDPI.

Minor revision:

- In the graph of Figure 4a, cytochalasine-D and 4c have no error value in all samples. The same is noted for the control group in Figure 3a.  

We thank the reviewer for pointing out this. We have added the error value to the control samples in the figures 3, 4 and 5. For figure 4 we have increased the number of samples to make n=3, and have added the error values in all samples.

 - The statistical method is not quite correctly chosen. The Mann-Whitney criterion is used to compare two independent small samples (n ≥ 3). In this case, it is better to change to a single-factor analysis of variance (ANOVA) or the Kraskel-Wallis criterion (checking the equality of medians).

We would like to thank the reviewer for addressing this point. As suggested by the reviewer, the “Mann-Whitney criterion is used to compare two independent small samples (n ≥ 3)”. In order to avoid confusion with the statistical method used in some data-graphs that were initially presented together and, because the biomolecules studied in this work have been treated as independent between them, we have changed the data representation on the figures 2, 3, 4 and 5 in the revised manuscript and have separated the original graph in single representations for each SL tested. We believe that this new data representation is more accurate and Mann-Whitney test will be correctly used for figure 3B, figure 4 and Figure 5B. Now, we show the differences in cilia length and cilia percentage of biomolecules-treated cells relative to control cells (DMSO).

- If the studied compounds affect the primary cilia, do they cause a change in the structure of microtubules, which are contained in large quantities in the primary cilia?

We thank the reviewer for raising this interesting point. In this work, using antibodies against acetylated a-tubulin, a microtubule marker, we observed by immunofluorescence microscopy that Sesquiterpene lactones can modify the primary cilium formation. Unfortunately, in our work, we have not studied the molecular mechanisms that cause this effect. We should use a different cell fixation method for immunofluorescence and make in vitro assays to study if the cilia defects observed are due to a change in the structure of microtubule dynamics.

In relation to this, it has been shown (10.1016/j.cbi.2003.10.008 ) that costunolide can interact with tubulin. In base of this work, we could think that cilia defects observed in our study might be directly related to changes in microtubule dynamics needed for cilia formation and elongation. Future studies will be aimed at studying this mechanism.

Reviewer 2 Report

Comments and Suggestions for Authors

The authors report on the effects of sesquiterpene lactones on primary cilia formation (ciliogenesis), in the title the authors are asked to indicate the type of sesquiterpene compound, e.g. guaiacolide type. The current title is too large and should be reduced.

The conformation of the compounds used in this study should be labelled.

The authors are asked to explain why the modified compounds have better activity. It is because the pharmacophores are all consistent.

The exact molecular mechanism needs to be further explored in the future.

Author Response

Reviewer 2:

We appreciate the comments of reviewer 2 and we will be supplying additional information in the revised manuscript to address the suggestions.

Comments and Suggestions for Authors

The authors report on the effects of sesquiterpene lactones on primary cilia formation (ciliogenesis), in the title the authors are asked to indicate the type of sesquiterpene compound, e.g., guaiacolide type. The current title is too large and should be reduced.

Although we agree with the referee that the current title could be more explicit about the type of sesquiterpene compound tested, not all of the products are guaianolides, and to name all the compounds will result in a title too long. In the same way, because we are discovering a new biological function for sesquiterpene lactones, should be more convenient to make a more general title.

The conformation of the compounds used in this study should be labelled.

The IUPAC names, with the indication of the stereochemistry of the different chiral centers, have been included in the manuscript on Materials and Methods, section 4.2- Semi-synthesis.

The authors are asked to explain why the modified compounds have better activity. It is because the pharmacophores are all consistent.

From a structural point of view, some structure-activity relationships have been described to explain why the modified compounds enhance the activity. In this type of compound, it is usual for the gamma-lactone group to be responsible for the activity. In all derivatives, this group is maintained in the structure. Therefore, the different activity must be due to other reasons, mainly related to the spatial conformation of the molecules. This study would have to be completed with molecular calculation studies to examine the influence of the compounds' conformation on the active center. Costunolide is a germacranolide, and the cyclos are eudesmanolides, so this different spatial arrangement may also influence the activity, not only the interaction with the pharmacophore.

The exact molecular mechanism needs to be further explored in the future.

We agree with the referee. Currently we are analyzing a RNAseq from cells treated with the products in order to get some clues about targets. Besides we are analyzing other signaling pathways.

Reviewer 3 Report

Comments and Suggestions for Authors

The paper is devoted to the influence of plant metabolites on the process of ciliogenesis. I have several questions and comments about this work.
Abstract
1)    «Their life cycle is linked to cell cycle, as cilia assemble in non-dividing cells of G0/G1 phases and disassemble with cell-cycle entry.»
In fact, primary cilia are present in cells also in the S phase and early G2 phase of the cell cycle, and are disassembled only before entering mitosis, and not the cell cycle, as written in the article.
2)    «Abnormalities in primary cilia structure or function are associated with developmental disorders (ciliopathies), heart disease and cancer.»
I think it is necessary to point out here that ciliopathies are associated not only with disturbances in the primary cilia, but also with disturbances in the functioning of the motile cilia of the ciliated epithelium in the respiratory system, oviducts, brain and sperm flagella.

3)    “The presence of cilium is regulated by cell cycle, as PC emerges on G0 (quiescent) or early G1 phase, and it is maintained until cells enter mitosis, being resorbed and absent in G2/M phases [5].”
G2 phase and mitosis should not be combined into one phase of the cell cycle. It is better to leave only mitosis here. Or use G2/M transition instead of G2/M phases.
The presence of cilium is regulated by cell cycle, as PC emerges on G0 (quiescent) or early G1 phase, and it is maintained until cells enter mitosis, being resorbed and absent in mitosis [5].
or
The presence of cilium is regulated by cell cycle, as PC emerges on G0 (quiescent) or early G1 phase, and it is maintained until cells enter mitosis, being resorbed during G2/M transition [5].

4)    Figure 3C. It is necessary to give enlarged images of the areas containing cilia on the right.
Legend to Figure 3. I did not find the (**) on the figure.

Figure 4C. and line 207-208
“As expected, we found that RPE cells treated with CytD during 18h (control) are arrested at G2/M.”

Flow cytometry results were interpreted incorrectly. Treatment of cells with cytochalasin for 18 hours leads to a block of cytotomy and the accumulation of tetraploid cells in the culture in the G1 phase (which are identical in the amount of DNA to cells in the G2 phase). These cells do not enter the S phase and this is associated with a general decrease in the proportion of cells in the S phase in the presence of cytochalasin. For correct analysis of a cell population, it is necessary to do microscopic study with cell labeling with BrDU or analogues to detect cells in the S phase.

5)    Figure 5C. It is necessary to give enlarged images of the areas containing cilia on the right.

Line 210
«... with similar percentage of cells arrested at G2 as control cells.»
Actin disassembly does not arrest cells in the G2 phase. A block of cytokinesis occurs after chromosome divergence in anaphase of mitosis. Next, two nuclei are formed, which sometimes merge into one. The authors just need to look at the cells after 1-2 hours of Cytochalasin treatment and see for themselves. Or study in more detail the literature on the effect of Cytochalasin D or Latrunculin A on mitosis and the cell cycle.

Questions:

The orientation of cilia relative to the cell culture plane may influence their length measurements. After all, not all eyelashes are oriented in the optimal way for measuring their length, parallel to the glass? How did the authors solve this problem?

  How accurately was the length of the cilia measured? The ideal option would be to use electron microscopy for analysis.

The fact that in actively dividing cells the percentage of primary cilia is lower than in resting cells does not mean the opposite pattern. That is, if you increase the proportion of cells with cilia, proliferation will slow down or stop.

In the Conclusion, the authors should explain in more detail how the development of drugs for ciliopathies is related to substances that reduce the percentage of cells with primary cilia? Ciliopathy is almost exclusively a genetically determined disease, and drug treatment, I think, is ineffective in this case.

Author Response

Reviewer 3:

We would like to thank the referee for taking the time to review our manuscript and for all the comments and suggestions which we believe improve the manuscript. We also want to thank the referee for the positive feed-back, especially for noticing the wrong interpretation of Figure 4.

Comments and Suggestions for Authors

The paper is devoted to the influence of plant metabolites on the process of ciliogenesis. I have several questions and comments about this work.
Abstract
1)    «Their life cycle is linked to cell cycle, as cilia assemble in non-dividing cells of G0/G1 phases and disassemble with cell-cycle entry. »
In fact, primary cilia are present in cells also in the S phase and early G2 phase of the cell cycle, and are disassembled only before entering mitosis, and not the cell cycle, as written in the article.

We thank the reviewer for this remark, which provides a more accurate description. We adapted the text accordingly in the Abstract and in line 127 (Results and Discussion; 2.1). “Their life cycle is linked to cell cycle, as cilia assemble in G0/G1 phases and disassemble before entering mitosis”.

2)    «Abnormalities in primary cilia structure or function are associated with developmental disorders (ciliopathies), heart disease and cancer. »
I think it is necessary to point out here that ciliopathies are associated not only with disturbances in the primary cilia, but also with disturbances in the functioning of the motile cilia of the ciliated epithelium in the respiratory system, oviducts, brain and sperm flagella.

We have considered the suggestion and added “Abnormalities in both primary cilia (non-motile cilia) and motile cilia (line 10, Abstract) and a new text in the Introduction: Page 2 line 47, 50-52.

3) “The presence of cilium is regulated by cell cycle, as PC emerges on G0 (quiescent) or early G1 phase, and it is maintained until cells enter mitosis, being resorbed and absent in G2/M phases [5].” G2 phase and mitosis should not be combined into one phase of the cell cycle. It is better to leave only mitosis here. Or use G2/M transition instead of G2/M phases.
The presence of cilium is regulated by cell cycle, as PC emerges on G0 (quiescent) or early G1 phase, and it is maintained until cells enter mitosis, being resorbed and absent in mitosis [5].
or
The presence of cilium is regulated by cell cycle, as PC emerges on G0 (quiescent) or early G1 phase, and it is maintained until cells enter mitosis, being resorbed during G2/M transition [5].

We have considered the suggestion and changed the text in the revised manuscript (line 35-36; line 127; 244-245).

4)    Figure 3C. It is necessary to give enlarged images of the areas containing cilia on the right.

We have considered the suggestion and we have added enlarged images from the merged in Figure 3A in the revised manuscript.

We would like to point out to the referee that, in the revised manuscript, the representation of the data has changed for a better understanding of the results. Now the Figure 3A is the old Figure 3C mentioned in this comment. We believe that the data are clearer now and show better the effect of each product independently.

Legend to Figure 3. I did not find the (**) on the figure.
Thank you for pointing this out. We have removed the (**) from the Figure 3 legend.

Figure 4C. and line 207-208
“As expected, we found that RPE cells treated with CytD during 18h (control) are arrested at G2/M.”

Flow cytometry results were interpreted incorrectly. Treatment of cells with cytochalasin for 18 hours leads to a block of cytotomy and the accumulation of tetraploid cells in the culture in the G1 phase (which are identical in the amount of DNA to cells in the G2 phase). These cells do not enter the S phase and this is associated with a general decrease in the proportion of cells in the S phase in the presence of cytochalasin. For correct analysis of a cell population, it is necessary to do microscopic study with cell labeling with BrDU or analogues to detect cells in the S phase.

We apologize for the wrong interpretation of the effect of Cytochalasin D on the cell cycle and thanks very much to the referee for pointing this out. We agree with the referee that Cytochalasin D treatment for 18 hours leads to a block of cytokinesis and it is totally true that we cannot differentiate in an accurate way the percentage of cell population in each cell cycle phase with a propidium-iodide (PI) assay. We only can affirm that treatment with the compound’s tested doesn´t change the pattern of CytD with our PI assay.

We have changed the interpretation of the results in the text (line 210-213) and in the legend to figure 4.  We also have changed the data-labels in Fig 4C, for a clearer interpretation of the data.

5)    Figure 5C. It is necessary to give enlarged images of the areas containing cilia on the right.
We have considered the suggestion and we have added enlarged images from the merged in Figure 5A in the revised manuscript.

Line 210
«... with similar percentage of cells arrested at G2 as control cells.»
Actin disassembly does not arrest cells in the G2 phase. A block of cytokinesis occurs after chromosome divergence in anaphase of mitosis. Next, two nuclei are formed, which sometimes merge into one. The authors just need to look at the cells after 1-2 hours of Cytochalasin treatment and see for themselves. Or study in more detail the literature on the effect of Cytochalasin D or Latrunculin A on mitosis and the cell cycle.

We agree with referee and because this comment is similar and related to the one mentioned above, the text has already been corrected with the changes that have been made for the previous comment, ((line 209-213) and in the figure 4 legend).

Questions:

The orientation of cilia relative to the cell culture plane may influence their length measurements. After all, not all eyelashes are oriented in the optimal way for measuring their length, parallel to the glass? How did the authors solve this problem?

How accurately was the length of the cilia measured? The ideal option would be to use electron microscopy for analysis.

To measure the length of the cilia in 2D cultures (cell cultures growing on coverslips) we used immunofluorescence microscopy using antibodies against acetylated a-tubulin as a ciliary marker and analyzed the maximum intensity projection of the acquired image. We use a semi-automatized software- ACDC, as mentioned in Materials and Methods for quantifying and measure cilia length. This is the most common analysis used. We agree with the referee that cilia orientation can influence the cilia length. To solve this problem, assuming that flat cilia and angle cilia represent the same population of cilia, we quantify a big number of cilia per condition (usually between 150-200 minimum) to minimize the number of cilia that are not well oriented. In the case of RPE cells, the most of the primary cilia are visible as flat cilia and are easy to measure in ImageJ or ACDC.

Electron microscopy is not commonly used for measuring the length of the cilia, although, it is true as the referee points out that should be the ideal option and the most accurate. This technique is most often used for structure analysis, e.g., to analyze the transition zone or the ciliary pocket. This restriction of use is mainly due to the high cost, time involved, and the scare number of cilia that can be quantify in the field.

The fact that in actively dividing cells the percentage of primary cilia is lower than in resting cells does not mean the opposite pattern. That is, if you increase the proportion of cells with cilia, proliferation will slow down or stop.

We agree with the referee that the relationship between cilia (number, length and signaling pathways) and the cell cycle remains still open and, before assuming the fact mentioned, further studies need to be done related to SLs. But, because it is known that cell cycle associated proteins can influence the ciliation process and, cilia and ciliary proteins can influence cell cycle progression, we might expect some effect on cell cycle process when increasing cilia formation in non-polarized cells under normal growth conditions.

In our work, we are suggesting that the effect provoked by the compounds on primary cilia formation is related directly to the ciliary pathways, and it is not an indirect effect of cell cycle dysfunction.

In the Conclusion, the authors should explain in more detail how the development of drugs for ciliopathies is related to substances that reduce the percentage of cells with primary cilia? Ciliopathy is almost exclusively a genetically determined disease, and drug treatment, I think, is ineffective in this case.

We thank the reviewer for raising this interesting point. We have changed the text in Conclusions to clarify the usefulness of our work in the discovery of new drugs to treat diseases with primary cilium dysfunction and/or tumors and the possibility to open new therapeutic avenues, mainly focused on diminished symptoms associated to the disease.

Round 2

Reviewer 3 Report

Comments and Suggestions for Authors

Dear Editor,

The authors have done a great job of correcting inaccuracies in the text and shortcomings in presentation. In some cases my views do not reflect the views of the authors, but in all cases I do not consider this a problem for publishing articles. Only further research will help resolve controversial issues. And the publication of this article will help attract the attention of researchers to this topic. I can conclude that the article in its present form can be published in the journal Toxins.